# How Anthropometrics of Young and Adolescent Swimmers Influence Stroking Parameters and Performance? A Systematic Review

**DOI:** 10.3390/ijerph19052543

**Published:** 2022-02-22

**Authors:** Miriam Alves, Diogo D. Carvalho, Ricardo J. Fernandes, João Paulo Vilas-Boas

**Affiliations:** Centre of Research, Education, Innovation and Intervention in Sport (CIFI2D), Porto Biomechanics Laboratory (LABIOMEP-UP), Faculty of Sport, University of Porto, 4099-002 Porto, Portugal; diogoduarte_03@hotmail.com (D.D.C.); ricfer@fade.up.pt (R.J.F.); jpvb@fade.up.pt (J.P.V.-B.)

**Keywords:** biomechanics, butterfly, backstroke, breaststroke, front crawl

## Abstract

The purpose of this systematic review was to investigate the relationship between anthropometric characteristics, biomechanical variables and performance in the conventional swimming techniques in young and adolescent swimmers. A database search from 1 January 2001 to 30 June 2021 was done according to the PRISMA statement, with 43 studies being selected for analysis. Those manuscripts were divided in butterfly, backstroke, breaststroke and front crawl techniques as main categories. The results showed the importance of the anthropometric variables for the performance of the young swimmer, although there was a lack of variables common to the studies that analysed the butterfly, backstroke and breaststroke techniques. For the front crawl technique there is a consensus among studies on the advantage of having higher height and arm span values, variables that concurrently with high body mass and lean body mass values, contribute positively to better stroke length and stoke index values.

## 1. Introduction

Swimming is an individual and cyclic sport that is influenced by a multifactorial group of determinants, from which the biomechanical and energetical factors seem to be the most relevant [1]. Swimming performance is determined by swimmers energetic profile, which is influenced by theirs biomechanical behaviour [1,2,3] that, in turn, is affected by individuals anthropometric characteristics [4]. This is known for adult and/or elite swimmers but it cannot be directly applied to younger counterparts since children and adolescents are not mini adults but individuals with specific characteristics and constraints [5,6,7].

As swimming velocity equals the product of stroke frequency (number of upper limbs cycles per unit of time) and stroke length (space travelled during a complete upper limbs cycle and is assessed by the velocity vs. stroke frequency ratio [8]), better understanding of the basic kinematical parameters behaviour and its relationship with velocity has been a major point of interest [2,9]. Another variable often explored is the stroke index (obtained by the product of stroke length and velocity) that is considered a valid indicator of swimming efficiency in adult [10] and young swimmers [7,11]. It is assumed that for a given velocity, swimmers moving at a higher stroke length have the most effective swimming technique and describes their ability to move at a given velocity with a lower stroke frequency [10].

Velocity increase or decrease happens due to a combined rise or reduction in stroke frequency and stroke length [8,9]. This relationship can be influenced by several variables, among which the anthropometric characteristics, with somatic attributes being largely inherited and determining swimming technique to a highest degree [5,6]. However, due to the growing process, young and adolescent individuals anthropometrical changes strongly influence these technical indexes and, consequently, performance [5,11,12]. It should be considered that anthropometric characteristics, among other factors that may influence the relationship between stroke frequency and stroke length (and affect swimming velocity), are related to stroke frequency and, more importantly, to stroke length in children [13,14]. This clearly demonstrates that anthropometric and biomechanical variables are associated and should be considered during young and adolescent swimmers performance monitoring.

In the last 20 years, it has been examined how anthropometric and other variables predict paediatric swimming performance e.g., [13,15,16] even if a lack of consistency in the range and type of variables examined was observed. For instance, in young boys 100 m, front crawl performance was predicted using upper limb length, horizontal jump and grip strength [13] but, for the same event, the anaerobic power, swimming index and critical velocity explained 88% of performance variability [15]. Strong associations have been found between anthropometric and biomechanical variables with swimming performance but with controversial results, hence there is a need to clarify the evidence in the four conventional swimming techniques. We aim to evaluate the published evidence and investigate the relationship between anthropometric characteristics, biomechanical variables and performance at butterfly, backstroke, breaststroke and front crawl in 9 to 18 years old swimmers. The following research questions were examined: (i) Are anthropometric variables related to butterfly, backstroke, breaststroke and front crawl performance? (ii) do anthropometric variables have a relationship with upper limbs cycle-related parameters?

## 2. Materials and Methods

Methodological procedures followed the standards for systematic reviews according to the Preferred Reporting Items for Systematic reviews and Meta-Analyses (PRISMA 312084) statement [17] and other relevant guidelines [18]. The EBSCOhost and Scopus databases were searched from 1 January 2001 to 30 June 2021 by two independent researchers that identified relevant studies using the combination of the “swimming*” and “young*” or “adolescents*” keywords with one of the “anthropometry*” and “biomechanics*” terms. Only cross-sectional and longitudinal experimental articles were considered and, after eliminating duplicates, results were screened according to the title and abstract (to exclude any irrelevant articles). Then, the full texts of potentially eligible studies were retrieved and independently evaluated for inclusion.

Studies including swimmers older than 18 years of age, undertaken in other scientific fields and/or sports, based on other swimming topics rather than anthropometrics and involving triathletes, divers or Paralympic swimmers, were excluded. Studies not relating anthropometric and biomechanical variables with performance, as well as those that only characterised or identified swimmers’ anthropometric profiles, were also excluded. Data relating to: (i) sample characteristics (number, sex and mean age); (ii) swimming technique and performance related distance; (iii) anthropometric variables; and (iv) major findings (anthropometric data that influenced biomechanical and performance related variables) were independently extracted. Concerning the research question, relevant studies were categorised in four main groups according to the swimming technique used to access performance (butterfly, backstroke, breaststroke and front crawl).

The methodological quality of the current study was assessed using the adapted Newcastle-Ottawa Scale [19] adopting these elements to evaluate the prospective studies risk of bias: (i) groups selection quality (sample representativeness and size, non-respondents and exposure ascertainment); (ii) groups comparability (study design or analysis and, confounding factors control) and (iii) outcomes assessment and statistical test. The maximum score on the Newcastle-Ottawa Scale is 10, evidencing the highest quality), being possible to attribute a maximum of five in the selection section, a maximum of two in comparability section and a maximum of three to the outcome of interest section [19].

## 3. Results

After extracting the manuscripts from the databases and removing duplicates, 7421 potentially relevant papers were screened, with the different steps of the selection process being displayed in Figure 1. After reviewing the study titles, abstracts and full text, 43 were included in the current systematic review and considered for further analysis. From these, the earliest one was published in 2005 and the most recent in January 2021, presenting a quality index of 7.28 ± 0.73 points (ranging from 6–9) (Table A1). Table 1 refers to the studies examined for each swimming technique, i.e., butterfly, backstroke, breaststroke and front crawl (with four, three, four and 38 studies, respectively). Since two studies were conducted on various swimming techniques that were included in several categories, hence the reason why the total number of papers did not match the sum of the categories of partial number. In addition, the study that evaluated the 200 m medley was included in the front crawl technique group.

From the included studies, front crawl was the most analysed swimming technique, and the 100 m event was the most studied (with >40% of the studies). The 50 and 400 m front crawl distances were also frequently studied (over than 40% of the studies), and some manuscripts focused on the 25 and 200 m front crawl. A single study conducted a 30 s front crawl tethered swimming test to calculate the propulsive force of the upper limb and two studies estimated the propulsive force of front crawl using the arm muscle area. In addition, one paper considered the European Swimming League (LEN) scores sum of the three best personal events and another used International Swimming Federation (FINA) points to access performance. However, only six studies were found that aimed to relate anthropometrical and biomechanical variables, and all of them focusing on the front crawl technique.

A comparison between the number of studies that analysed a certain variable and the ones that found a relation between the variables and performance was made for each swimming technique (see Figure 2). Swimmers’ height was assessed on three of the butterfly included studies and only one reported a relationship with performance. In the backstroke related studies, no common variable was evaluated by the three studies, but sitting height, forearm girth and arm relaxed girth were accessed in two papers and were associated with performance. In the four breaststroke studies, there was also no variable evaluated commonly. Lastly, arm span, height and body mass were the most evaluated variables among the front crawl-related studies.

## 4. Discussion

We aimed to investigate the association between anthropometric characteristics, biomechanical variables and swimming performance at different swimming techniques in young and adolescent swimmers. The current study revealed a lack of variables common to the studies that analysed the butterfly, backstroke and breaststroke techniques. For front crawl, arm span and height were the commonly observed variables related to performance. In addition, in front crawl, the anthropometric characteristics were associated with biomechanical variables.

Arm span was one of the variables that was positively related to performance in all four swimming techniques and in individual medley (r = 0.3–0.9). However, findings demonstrate the importance of having shorter forearm length for better butterfly, backstroke and breaststroke performances [16,52]. Longer extremities allow the swimmer to perform fewer upper limbs cycles for the same distance [57,58] and helps to achieve a higher moment of force (propulsive and resistive) in a single upper limb cycle. In addition, swimmers that have long body segments develop a greater propulsive force as opposed to resistive forces to advancement [59]. On the other hand, longer lever lengths (like the forearm length) could also be mechanically disadvantageous since the involved muscles have to exert greater force and energy when the same drag is associated to a longer lever arm length. 

A consistent positive relationship with height and front crawl was found (r = 0.3–0.9). This could be explained by the fact that a longer body has more streamline properties since the zone of boundary layer separation may be relatively more caudal due to slenderness effect on the Reynolds number, which generates a smaller wake and a reduced wave compared to smaller bodies [60]. On the other hand, the lack of relationship between height and performance can be explained by the fact that the advantage of longer levers was mainly limb-specific rather than a more general whole-body advantage [35] or/and that if muscle mass does not follow a body height increase, the increase in the length is not of great benefit to the swimmer as he is not able to use height (longer levers) to achieve a higher moment of force.

Biacromial and bi-iliac breadths were variables related to performance in all four techniques (r = 0.4–0.7). When a body moves in the liquid environment, a stagnation of flow occurs at its anterior extremity and the pressure drag a swimmer is subjected to is higher at those regions. These occur mainly around the body where there are sudden changes in shape, such as the shoulders, hips and knees [61]. A swimmer with high biacromial breadth values (broad shoulders) and low bi-iliac breadths values (narrow hips) will have a lower drag coefficient [62], as the human body adopts a more hydrodynamic position the closer it is to the form of a drop of water [63]. It was observed that bi-iliac breadths made a positive correlation with performance, which is probably because swimmers with wide shoulders will have a hip proportional to their size, i.e., high values of bi-iliac breadths. Thus, since high bi-iliac breadths values can mean also high biacromial breadth values, the biacromial/bi-iliac diameter index suggested by Clarys [63] should be taken into consideration.

The arm span/height index suggests a hydrodynamic advantage since it is achieved through high values of biacromial diameter and upper limb length in detriment of torso and lower limbs lengths [63]. However, no association was found between this index and performance in the studies included in the current review. Conversely, sitting height made a positive effect in backstroke and front crawl technique performance (r value: 0.5 to 0.7). Since the height of a swimmer is proportional to torso values, the taller a swimmer is, the higher sitting height values are. Thus, the relationship between sitting height and performance can be explained by the fact that swimmers with longer torso values are also taller. In this way, taller swimmers show a decrease in the Froude number and in wave-making resistance, which allows them to cut the water with less resistance and their long bodies give them an automatic edge [64,65].

Advantages in having greater hand width, hand and foot surface areas and lengths for breaststroke, front crawl performance and 200 m medley were reported among studies. Additionally, a positive relationship between hand surface area and front crawl thrust was observed [55]. Upper body and arm lengths also presented a positive association with breaststroke performance, and upper and lower limbs and thigh lengths with front crawl performance (r = 0.2–0.7). These could be mainly due to propulsive force and, hence, swimming performance being positively affected by higher propulsive surface areas. This may increase hydrodynamic lift force to propel the swimmer through the water and allow him to perform fewer upper limbs cycles for the same distance [66].

Having greater limb segment-length ratios (arm length ratio = forearm length/arm length; foot by leg ratio = foot length/leg length) seems to be an advantage for front crawl performance [35]. The negative association between leg and performance may be explained by the fact that longer legs in swimming may alter the flotation of the swimmer, potentially resulting in a sinking of the lower limbs. An increase in the downward inclination of the legs would increase pressure drag, due to an increase in immerged body surface and cross-sectional area of the body of the swimmer and increasing the energy cost of swimming. A greater foot-to-leg ratio, with a greater foot size and a shorter leg length to reduce the downward inclination of longer legs may reduce drag. Conversely, results illustrated that leg length made a positive contribution to backstroke and breaststroke performance [48].

Body mass was one of the common anthropometrics studied in front crawl technique and associations were only found in this swimming technique as well as in the 200 m individual medley (r = 0.3–0.9). In athletes, body mass can be an indicator of the active muscle mass. Consequently, an increase of body mass may be related to a higher muscle mass. However, an increase in body mass can also be an indicator of high values of body fat mass. Muscle mass can represent body strength of the swimmer; if these values are low, a decrease in swimming performance may be expected. It should be considered that body mass does not identify and represent lean body mass or body fat mass proportions and so, for a more detailed evaluation, these variables should be measured independently.

Moreover, results demonstrated that fat mass was the only whole body-size characteristic negatively associated with all four techniques velocity [16,52]. The disadvantage of having higher fat mass suggests that swimmers require greater lean body mass and greater muscle strength to propel themselves faster through the water. In parallel, higher fat mass may impose increased values of body cross-sectional area and, consequently, total drag. Regarding more detailed body composition variables, it seems that total sum of skinfolds [32] and biceps skinfold [41,47] compromised front crawl performance. Skinfolds measures are used to assess the skinfold thickness, so that a prediction of the total amount of body fat of the swimmer can be made. Therefore, skinfold measurements reflect adiposity values. Swimmers with a more developed muscular system, have lower skinfold values and, therefore, may have better performances, which can explain the fact that skinfolds impaired performance.

Lean body mass was only related to performance in 200 m breaststroke and in front crawl techniques (r = 0.4–0.8) [5,31,35]. A larger lean body mass and, thus, a greater muscle mass, could positively influence biomechanical values by enhancing the force applied in each upper limbs cycle and the capacity to maintain good SI under exhaustion conditions [67]. Positive associations between biceps circumference in contraction with breaststroke and front crawl performance were also found [47]. This variable is related to muscle mass, thus the higher it is the higher may be the strength of the upper limbs and, possibly, propulsion generation. 

Associations between arm relaxed and forearm girths and performance were found for all four techniques, in which swimming speed was negatively influenced by arm relaxed girth (r = 0.2–0.4) [52]. Authors suggested that the arm girth ratio (forearm girth/relaxed arm girth) was possibly reflecting a measure of muscle strength. Similarly, findings demonstrated that an increase in forearm girth improves breaststroke swimming performance and that having a greater wrist girth impairs performance [42]. A possible explanation can be that a large wrist girth would increase hydrodynamic drag, therefore increasing the energy cost of swimming, and a greater forearm girth seems to generate higher propulsive force and consequently an easier propulsion through water. In addition, an advantage of having greater calf girth and lower values of ankle girth was reported for butterfly swimming [16,52]. The ratio calf girth/ankle girth seems to reflect the greater muscle strength in the legs associated with faster butterfly swimmers.

Moreover, it was mentioned that chest circumference negatively influenced the 100 m front crawl male performance (r = 0.6) [13]. Swimming velocity is determined by the propulsive force and the hydrodynamic drag force. There are certain resistive segments which should preferably be of small dimensions (e.g., chest circumference). These segments should present the referred characteristics because the hydrodynamic lift drag forces depend on the cross-sectional area of the swimmer’s body considering the direction of its displacement. Chest circumference seems to represent part of body cross-sectional area of the swimmer meaning that an increased chest circumference may increase drag values of the swimmer, which may impair performance.

Only six studies investigated associations between upper limbs action and anthropometrical variables, and all of them only focused on the front crawl technique. An advantage in having a greater arm span and height for higher values of stroke length and SI were reported (r = 0.4–0.8) [5,6,11,21,25,28]. Likewise, body mass and lean body mass were positively associated with stroke length and stroke index (r = 0.5–0.7) [5,6,11,28]. Stroke length and stroke index were also positively related to upper and lower limb lengths in all distances (100, 200 and 400 m) [28].

There is a consistent association between arm span, height, upper and lower limb lengths and stroke length. It seems that swimmers with the highest height have higher arm span and surface areas as well. Once the linear distance covered per revolution by a given landmark during angular motion is a function of the angular displacement and its linear distance to the fulcrum, in swimmers, longer extremities allow them to perform fewer upper limbs cycles for the same distance [57,58]. In addition, these surface areas, when properly oriented, induce an increase in propulsive forces. By consequence, such swimmers can achieve a higher stroke length and consequently a higher velocity [2,3]. A few studies also found an association between body mass, lean body mass and upper limbs cycle-related parameters, being observed that arm muscle area and lean body mass were positively associated with arm propulsive force (tethered swimming) in front crawl [56].

The current study showed interesting results, yet there are limitations that should be considered and addressed in future research. The swimming distance was not considered when comparing the studies (due to the small number of studies in three techniques), as well as maturational factors and years of experience were not considered.

## 5. Conclusions

In conclusion, the included studies about the association between anthropometrical and biomechanical variables and performance were in larger number for the front crawl technique followed by breaststroke, backstroke and butterfly, all with expressive fewer analysis. The lack of common measures between studies in butterfly, backstroke and breaststroke swimming techniques does not allow us to state which are the main anthropometric performance determining variables for these swimming techniques. However, anthropometric variables seem to be important for performance, particularly during growth, and there is a need for further study on this topic. For the front crawl technique, there seems to be a consensus among studies on the advantage of having higher values of height and arm span. Associations between anthropometrical characteristics and upper limbs cycle parameters were only found in front crawl. Stroke length and stroke index seem to benefit from higher values of height and arm span. In addition, higher values of body mass and lean body mass also appear to improve these upper limbs action variables.

## Figures and Tables

**Figure 1 ijerph-19-02543-f001:**
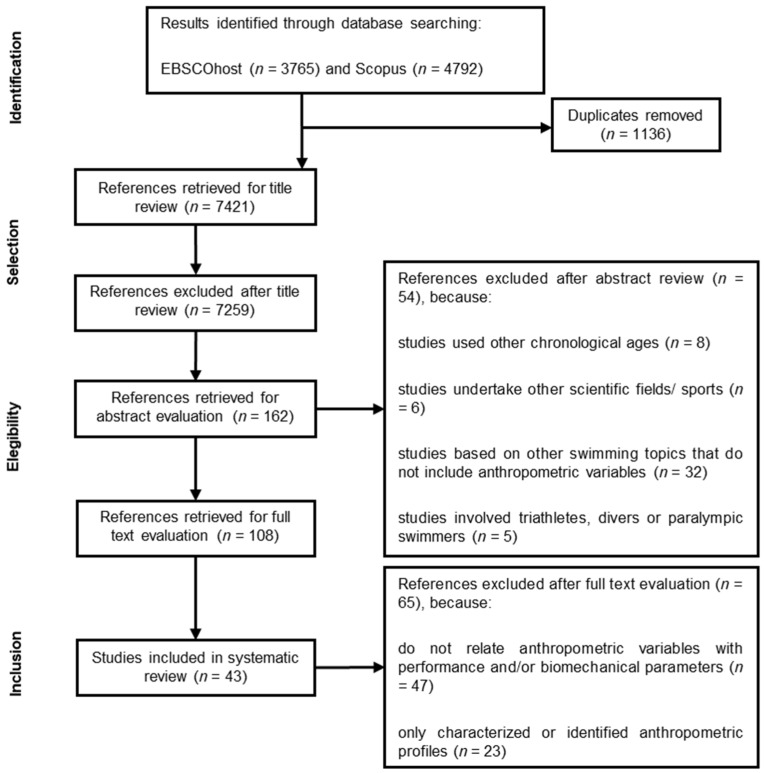
PRISMA flowchart for the current study.

**Figure 2 ijerph-19-02543-f002:**
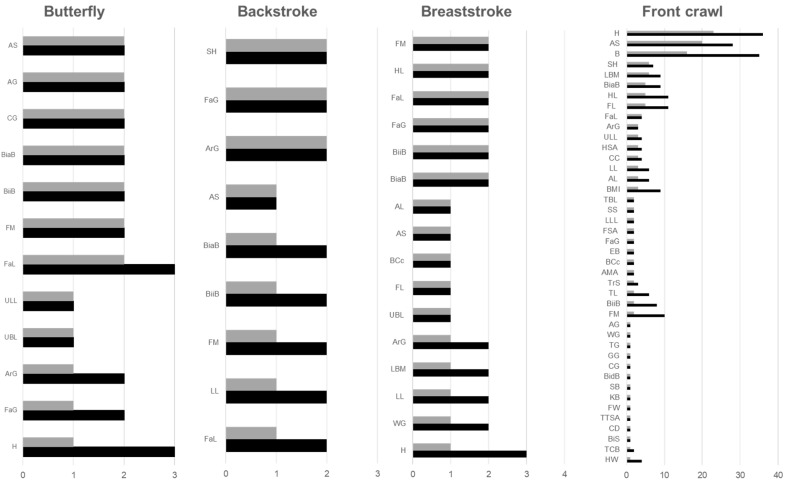
Frequency chart of anthropometric variables (black bars) and their relationship with performance (grey bars) of the four swimming techniques in a descending order of consistent relationship. ankle girth (AG), arm length (AL), arm muscle area (AMA), arm relaxed girth (ArG), arm span (AS), biceps circumference in contraction (BCc), biacromial breadth (BiaB), bi-deltoid breadth (BidB), bi-iliac breadth (BiiB), biceps skinfold (BiS), body mass (BM), body mass index (BMI), calf girth (CG), chest circumference (CC), chest depth (CD), elbow breadth (EB), forearm girth (FaG), forearm length (FaL), foot length (FL), fat mass (FM), foot surface area (FSA), foot width (FW), gluteal girth (GG), body height (H), hand length (HL), hand surface area (HSA), knee breadth (KB), lean body mass (LBM), leg length (LL), lower limb length (LLL), sitting height (SH), styloid breadth (SB), sum of skinfolds (SS), total body length (TBL), thigh girth (TG), thigh length (TL), transverse chest breadth (TCB), triceps skinfold (TrS), trunk transverse surface area (TTSA), upper body length (UBL), upper limb length (ULL), wrist girth (WG).

**Table 1 ijerph-19-02543-t001:** Characteristics of the included studies regarding anthropometry and swimming performance.

Author(s)	Swimmers Number, Sex and Age	Swimming Event	Anthropometric Variables	Major Findings
Geladas et al. (2005) [13]	178 males, 85 females (12.8 ± 0.1 and 12.7 ± 0.1 yo)	100 m front crawl (s)	H, BM, ULL, HL, FL, CC, BiaB, BiiB, FM	H (r = −0.31), ULL (r = −0.23) and HL (r = −0.30) were predictors of performance, in girls
				H (r = −0.61), BM (r = −0.65), ULL (r = −0.64), HL (r = −0.57), FL (r = −0.49), CC (r = −0.64), BiaB (r = −0.61) and BiiB (r = −0.46) were predictors of performance, in boys
Jürimäe et al., (2007) [11]	29 males (15 prepubertal and 14 pubertal: 11.9 ± 0.3 and 14.3 ± 1.4 yo)	400 m front crawl (s)	H, BM, AS, BMI, FM, LBM	H (r = –0.658), BM (r = –0.620), BMI (r = –0.479) and AS (r = –0.688) were related to performance
				SL and SI were related to H (r = 0.707; r = 0.721), BM (r = 0.693; r = 0.714), BMI (r = 0.562; r = 0.582), LBM (r = 0.689; r = 0.690) and AS (r = 0.746; r = 0.758)
Sekulić et al., (2007) [20]	68 (15.2 ± 3 yo)	50 and 400 m front crawl (m/s)	H, BM, BMI	H (0.56 < r < 0.72), BM (0.44 < r < 0.72) and BMI (0.39 < r < 0.60) were related with performance (except male BMI with 400 m performance)
Lätt et al., (2009a) [5]	26 females (1st, 2nd, 3rd measurement: 12.7 ± 2.2, 13.6 ± 1.9 and 14.6 ± 1.9 yo)	400 m front crawl (s)	H, BM, AS, BMI, FM, LBM	LBM (R^2^ = 0.318) was a predictor of performance
				SL was related to H (r = 0.411) and SI was related to H (r = 0.460) and AS (r = 0.413)
Lätt et al., (2009b) [6]	29 males (1st, 2nd, 3^rd^ measurement: 13.0 ± 1.8, 14.0 ± 1.8 and 15.1 ± 1.8 yo)	400 m front crawl (s)	H, BM, AS, BMI, FM, LBM	Performance was related to H (r = −0.47) and AS (r = −0.40). AS (R^2^ = 0.454) was a predictor of performance
				H (r > 0.40) and AS (r > 0.39) were related to v, SL and SI. BM was related to v (r = 0.45) and SI (r = 0.46)
Strzała & Tyka (2009) [21]	26 (16.1 ± 1.1 yo)	25 and 100 m front crawl (m/s)	BM, H, AS, TBL, LBM	TBL (r = 0.58) and AS (r = 0.55) were related to 25 m performance. TBL (r = 0.61) was related to 100 m performance
				TBL (r = 0.54; r = 0.61), AS (r = 0.51; r = 0.58) and LBM (r = 0.47; r = 0.57) were related to SL in 25 m and 100 m
De Mello Vito & Bohme (2010) [15]	24 males (13.0 ± 0.7 yo)	100 m front crawl (m/s)	BM, H, AS, HL, HW, FL, FW, BiaB, BiiB, AS/H index, BiaB/BiiB index, FM	BM (r = 0.59), BiaB (r = 0.57) and H (r = 0.53) were related with performance
Lätt et al., (2010) [22]	25 males (15.2 ± 1.9 yo)	100 m front crawl (s)	H, BM, BMI, FM, LBM	Performance was related with H (r = −0.536) and AS (r = −0.557). AS (R^2^ =0.485) was a predictor of performance
Saavedra et al., (2010) [23]	66 males, 67 females (13.6 ± 0.6 and 11.5 ± 0.6 yo)	100, 200, 400, 800 or 1500 m (all swimming techniques) (sum of LEN scores in the 3 best personal events)	H, SH, AS, BM, HL, HW, FL, FW, BiaB, BiiB, BitB, KB, EB, WB, CC, BCc, GG, TG, LG, AS/H index, BiaB/H index, CC/H index, GG/H index, BMI, SS, FM	SH (r = 0.579) was related to male performance
Maszcyk et al., (2012) [24]	189 (12.0 ± 0.5 yo)	50 m and 800 m front crawl (s)	H, BM, HL, AS, FL	50 m FL (B = 0.90) and H (B = −0.74); 800 m HL (B = 0.34) entered regression models to predict time
Morais et al., (2012) [25]	73 males, 41 females (12.7 ± 1.0 and 11.5 ± 0.7 yo)	100 m front crawl (s)	AS, HSA	Performance was related with AS (r = −0.35). SI was related with SL and AS
Mezzaroba et al., (2013) [26]	17 males, 16 females (13.6 ± 2.4 and 13.2 ± 2.3 yo)	100, 200 and 400 m front crawl (m/s)	BM, H, ULL, LLL, LBM	In boys, H and LLL were predictors of 100 and 200 m performance and BM was a predictor of 400 m performance. In girls, H was a predictor of all distance performances. LBM and ULL were predictors for all distances
Morais et al., (2013) [27]	15 males, 18 females (12.30 ± 0.63 and 11.77 ± 0.92 yo)	100 m front-crawl (s)	BM, H, AS, CC, TTSA, HSA, FSA	BM (0.96 < r < 0.99), H (0.97 < r < 0.99), AS (0.93 < r < 0.97), CC (0.94 < r < 0.96), HSA (0.92 < r < 0.96), FSA (0.78 < r < 0.96) and TTSA (0.49 < r < 0.79) were related to front-crawl performance.
Mezzaroba et al., (2014) [28]	46 males (10.7 ± 0.9, 13.0 ± 0.5, 15.3 ± 0.5 and 17.0 ± 0.7 yo)	100, 200 and 400 m front crawl	BM, FM, H, ULL, LLL	ULL was a predictor of SL and SI in all distances. H was a predictor of SI in all distances. LLL was a predictor of SF in 200 and 400 m
Moreira et al., (2014) [29]	12 males, 13 females (12.8 ± 0.9 and 12.0 ± 0.9 yo)	25 m front crawl (m/s)	H, AS, HSA, FSA	Only in the second test, H (r = 0.72), AS (r = 0.69), HSA (r = 0.72) and FSA (r = 0.59) were related to performance
Nasirzade et al., (2014) [30]	23 males (13.9 ± 0.9 yo)	50 m front crawl (s)	H, BM, BMI, AS, BiaB, AL, TL, LL	Performance was related with H (r = −0.43) and AS (r = −0.50)
Strzała et al., (2014) [31]	27 males (15.7 ± 2.0 yo)	200 m breaststroke (m/s)	LBM	LBM (r = 0.38) was related to 200 m turns velocity
Bond et al., (2015) [32]	21 males, 29 females (13.6 ± 1.7 and 13.4 ± 1.3 yo)	100 m front crawl (s)	H, BM, SS, AL, FaL, HL, TL, LL, FL	H (r = −0.654), BM (r = −0.543), SS (r = 0.410), AL (r = −0.561), FaL (r = −0.483), HL (r = −0.626), TL (r = −0.350) and FL (r = −0.494) were related with performance
				SS, TL, LL, HL and H were predictors of performance
Cochrane et al., (2015) [33]	30 male (12.4 ± 2.7 yo)	Front crawl EPF (kgf)	BM, H, FM, LBM, TrS, AMA, ArG	BM (r = 0.77), H (r = 0.84), LBM (r = 0.93), ArG (r = 0.95) and AMA (r = 1.00) were related to estimated propulsive force
Nasirzade et al., (2015) [34]	22 males (14.52 ± 0.77 yo)	200 m front crawl (s)	H, BM, AS, BiaB, TL, LL, AL	H (r = −0.71), AS (r = −0.62), BiaB (r = −0.48) and AL (r = −0.44) were related to performance.
Nevill et al., (2015) [35]	50 (13.5 ± 1.5 yo)	100 m front crawl (m/s)	H, BM, FM, LBM, AL, FaL, HL, TL, LL, FL	Performance was related with LBM, FaL, AL, FL and LL
Figueiredo et al., (2016) [36]	103 (11.8 ± 0.8 yo)	25 m front crawl (m/s)	BM, H, AS, HL, HW, FL, FW, SL/AS	HW was related with performance
Morais et al., (2016) [37]	49 males, 51 females (12.5 ± 0.76 and 12.2 ± 0.71 yo)	100 m front-crawl (s)	BM, H, AS	None of the anthropometric variables entered the final model to predict performance.
Akşit et al., (2017) [38]	25 males, 25 females (12.4 ± 1.2 and 12.0 ± 0.9 yo)	200 m and 400 m front crawl Critv(m/s) and EPF (kgf)	BM, H, BMI, SS, SH, AS, AL, FaL, TL, LL, LLL, HL, FL, BiaB, BiiB, TCB, FW, EB, KB, SB, BidB, ArG, BCc, CC, CG, FaG, GG, TG, WG, AG	Anthropometric characteristics were related (r) to Critv (19 variables, ranging from 0.34 to 0.66 and 27 variables, ranging from 0.37 to 0.81, for females and males, respectively), and with EPF (24 variables, ranging from 0.38 to 0.87 and 26 variables, ranging from 0.39 and 0.90, for females and males, respectively)
Morais et al., (2017) [39]	44 males, 47 females (12.0 ± 0.8 and 11.2 ± 1.0 yo)	100 m front crawl (s)	BM, H, AS	AS was a predictor of performance
Sammoud et al., (2017) [16]	103 males, 64 females (13.1 ± 2.8 and 13.6 ± 2.6 yo)	100 m butterfly (m/s)	BM, H, SH, AS, FM, LBM, BMI, ULL, AL, FaL, HL, LLL, TL, LL, FL, ArG, FaG, WG, TG, CG, AG, BiaB, BiiB	FM (B = −0.011), FaL (B = −0.356), AS (B = 0.428), BiaB (B = 0.489), BiiB (B = 0.292) CG (B = 0.573) and AG (B = −0.412) were predictors of performance
Mitchell et al., (2018) [40]	22 males, 26 females (16.5 ± 1.2 and 15.5 ± 1.1 yo)	100 and 200 m front-crawl (s)	H, SH, BM, Sk, FL, BiaB, BiiB, CD, TCB, EB	BM and EB were predictors of 100 m male performance and BM for 200 m. CD and SH were predictors for 100 m female performance and CD for 200 m.
Rozi et al., (2018) [41]	25 males (15 ± 1.5 yo)	100 m front crawl (s)	H, BM, SH, AS, BiaB, BiiB, CD, CC, ULL, Sk, BiS, TrS	Performance was related with H (r = 0.810), BM (r = 0.720), SH (r = 0.762), AS (r = 0.835), BiaB (r = 0.751), BiiB (r = 0.608), CD (r = 0.345), CC (r = 0.720), ULL (r = 0.784) and Sk (0.405< r < 0.666). Predictors of performance were AS, TrS, BiiB and BiaB
Sammoud et al., (2018) [42]	39 males, 20 females (11.5 ± 1.3 and 12.1 ± 1.0 yo)	100 m breaststroke (m/s)	BM, H, AS, SH, FM, LBM, BMI, ULL, AL, FaL, HL, LLL, TL, LL, FL, ArG, FaG, WG, TG, CG, AG, BiaB, BiiB	FM (B = −0.018), FaL (B = −0.418), HL (B = 0.309), LL (B = 0.673), BiaB (B = 0.565), BiiB (B = 0.403), FaG (B = 0.690) and WG (B = −0.348) were predictors of performance
Silva et al., (2018) [43]	23 males, 26 females (15.7 ± 0.8 and 14.5 ± 0.8 yo)	50 m front crawl (m/s)	H, AS, BM	H (r = 0.42) was related to male performance
Bielec et al., (2019) [44]	26 males, 15 females (12.1 ± 0.5 and 12.2 ± 0.5 yo).	50 m front crawl; 200 m IM (FINA points)	H, HW, HL, AS, BM, BMI, FM	H (r = 0.60; r = 0.67), AS (r = 0.57; r = 0.60) and HL (r = 0.52; r = 0.51) were related with front crawl performance (in boys and girls, respectively). In boys, H (r = 0.57), AS (r = 0.49), HL (r = 0.44) and FM (r = −0.56) were related with IM performance
Demirkan et al., (2019) [45]	10 males and 12 females (11.9 ± 0.7 and 12.1 ± 0.9 yo)	50 m butterfly, backstroke, breaststroke and front crawl (s)	H, UBL, ULL, HL, FL, BCc	Butterfly performance was related with H and ULL. ULL and UBL were predictors of butterfly performance
Ferreira et al., (2019) [46]	14 females, 29 males (10.74 ± 0.91 and 11.90 ± 1.08 yo)	400 m front crawl (m/s)	BM, H	BM (0.34 < r < 0.38) and H (0.43 < r < 0.48) were related to performance
Rozi et al., (2019) [47]	30 males, 21 females (15.1 ± 1.6 and 14.5 ± 1.5 yo)	100 m front crawl (s)	H, BM, SH, AS, BiaB, BiiB, CD, CC, BCr, BCc, ULL, Sk, BiS, TrS	H, AS and ULL were related with performance. AS, BiS and BCc were predictors of male performance. SH was a predictor of female performance. TrS and BCc were predictors of performance of 13–15 years old
Sammoud et al., (2019) [48]	30 males, 33 females (14.0 ± 0.6 and 13.0 ± 1.2 yo)	100 m backstroke (m/s)	BM, H, SH, FM, LBM, BMI, AL, FaL, HL, TL, LL, FL, ArG, FaG, TG, CG, BiaB, BiiB	SH (B = 0.833), LL (B = 0.258), FaG (B = 0.519) and ArG (B = −0.627) were predictors of performance
Strzała & Tyka (2019) [49]	15 (17.3 ± 0.59 yo)	50 m front crawl (m/s)	BM, LBM, H, TBL, AS	BM (r = 0.63), LBM (r = 0.78), H(r = 0.55), TBL (r = 0.58) and AS (r = 0.52) were related to front-crawl performance.
Ferraz et al., (2020) [50]	98 (12.63 ± 0.76 yo)	50 m and 400 m front crawl (s)	BM, H, AS	Associations were found between H (r = −0.553; r = −0.577), BM (r = −0.450; r = −0.434) and AS (r = −0.477; r = −0.500) with 50 m and 400 m time
Morais et al., (2020) [51]	12 males, 6 females (15.81 ± 1.62 yo)	25 m front crawl	BM, H, AS, ULL, HSA	Propulsive force presented a direct and positive relationship with HSA
Nevill et al., (2020) [52]	202 males (11.5 ± 1.3, 13.1 ± 2.8, 14.0 ± 0.6 and 19.0 ± 3.8 yo)	100 m butterfly, backstroke, breaststroke and front crawl (m/s)	BM, H, AS, FM, SH, ULL, AL, FaL, HL, LLL, TL, LL, FL, ArG, FaG, WG, TG, CG, AG, BiaB, BiiB	Seven predictor variables common to all techniques: FM (B = −0.089), FaL (B = −0.247), ArG (B = −0.272), FaG (B = 0.409), BiaB (B = 0.434), BiiB (B = 0.171), AS (B = 0.327)
	161 females (12.1 ± 1.0, 13.6 ± 2.6, 13.0 ± 1.2 and 15.9 ± 2.7 yo)			CG (B = 0.689) and AG (B = −0.526) were predictors of butterfly performance. SH (B = 0.492) was a predictor of backstroke performance
Zacca et al., (2020) [53]	10 males, 14 females (14.9 ± 1.0 and 14.2 ± 0.8 yo)	400 m front crawl (s)	H, BM, AS, BMI	The relative contributions of anthropometric variables for performance ranged between 7 and 19%.
Ferreira et al., (2021) [54]	10 females, 24 males (11.24 ± 0.88 and 12.51 ± 0.99 yo)	400 m front crawl (m/s)	BM, H	BM (r = 0.35) and H (0.39 < r < 0.41) were related to performance
Morais et al., (2021) [55]	10 males, 10 females (15.40 ± 0.30 and 14.43 ± 0.23 yo)	25 m butterfly (m/s)	AL, FaL, HSA	Anthropometric variables did not enter final model to predict butterfly performance
Oliveira et al., (2021) [56]	53 males, 75 females (13.6 ± 1.8 and 12.5 ± 1.8 yo)	30 s front crawl tethered (propulsive force of the arm)	BM, H, AS, AMA, FM, LBM.	BM (r = 0.29), H (r = 0.25), AS (r = 0.30), AMA (r = 0.28) and LBM (r = 0.42) were related with arm propulsive force

Ankle girth (AG), arm length (AL), arm muscle area (AMA), arm relaxed girth (ArG), arm span (AS), unstandardised Beta (B), biceps circumference in contraction (BCc), biceps circumference relaxed (BCr), biacromial breadth (BiaB), bi-deltoid breadth (BidB), bi-iliac breadth (BiiB), bi-trochanteric breadth (BitB), biceps skinfold (BiS), body mass (BM), body mass index (BMI), calf girth (CG), chest circumference (CC), chest depth (CD), critical velocity (Critv), elbow breadth (EB), estimated propulsive force (EPF), forearm girth (FaG), forearm length (FaL), foot length (FL), fat mass (FM), foot surface area (FSA), foot width (FW), gluteal girth (GG), body height (H), hand length (HL), hand surface area (HSA), hand width (HW), individual medley (IM), knee breadth (KB), lean body mass (LBM), leg length (LL), lower limb length (LLL), leg circumference (LG), pearson’s correlation (r), sitting height (SH), skinfolds (Sk), styloid breadth (SB), sum of skinfolds (SS), total body length (TBL), thigh girth (TG), thigh length (TL), transverse chest breadth (TCB), triceps skinfold (TrS), trunk transverse surface area (TTSA), upper body length (UBL), upper limb length (ULL), velocity (v), wrist breadth (WB), wrist girth (WG).

## Data Availability

Not applicable.

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
