# Peer review of "How Anthropometrics of Young and Adolescent Swimmers Influence Stroking Parameters and Performance? A Systematic Review"

_ijerph, 2022, doi:10.3390/ijerph19052543_

Round 1

Reviewer 1 Report

First of all, I am not English native speaker, so I hope you understand me well.

The authors did a good job with this study.

However, there are some things that need to be explained better or corrected.

Line 10-20: I think that it is necessary to write the date of search the articles in the abstract.

 Line 21: Change the keywords repeated in the title.

Line 42-53: I think that the decrease of velocity will be influenced by the physiological and mechanical fatigue. Perhaps this should be included in this paragraph in 2-3 sentences. Relationship between fatigue and biomechanical changes.

Line 70: Why do you use a range of twenty years? It is possible that you loss some relevant articles.

Line 66-95: Good job with the methods!

Line 101: Something wrong with a word. Please, correct it.

Line 128-129: Please, explain LEN and FINA abbreviations for the no experts in swimming.

Line 141: Maybe it is necessary to explain better this figure. I don’t understand the numbers of the x axis for each swimming technique. For example, Butterfly and AS: both bars in the number 2.

Line 162-171: Do you have some numbers to know the relationship between performance and biomechanical variables? The same for the following paragraphs.

Reviewer 2 Report

The authors present a manuscript with a very interesting topic, I have really enjoyed reading it. However, there are a number of aspects that should be fixed before publication.

The title seems well thought out and very clear.

In the abstract, as well as in the aim of the introduction, the authors should change the objective, since a systematic review does not have the objective of summarizing, but rather of drawing conclusions and answering research questions. In fact, in the first paragraph of the discussion, the authors write the objective in this way. I take this opportunity to ask the authors to include the research question(s) at the end of the introduction section. The title already offers an advance to the reader. IDEM for L62

In the keywords authors should not be repeated with words from the title, the authors of a systematic review like this one should already know this.

There are statements that have miss the reference in the introduction (L26, L45)

It could be that the word "factors" is missing in L26 (e.g. biomechanical and energetical factors?)

It confuses me a bit to read a systematic review on the relationship of performance in young people with their anthropometric measurements and that the authors include the following statement (L28): "This is known for adult and/or elite swimmers but it cannot be directly applied to younger counterparts since children and adolescents are not mini adults but individuals with specific characteristics and constraints."

This leads us to think that the objective of this review may lack the analysis of other factors (relative age, maturation, years of experience, etc.). Perhaps the authors should consider it in the limitations section.

Missing parentheses (L34?)

Method

Authors should include more databases in their search for a Q1 journal publication. WoS, Sage Journals, ScienceDirect, Springerlink, Taylor and Francis Online, Wiley, WorldCat, etc. Otherwise, an important bias in the methodology is being assumed.

I don't understand why the authors add " * " to the search word. In any case, * is used to truncate words, I could understand if the authors truncated Swim* because articles containing "swimming" or "swimmers" would be listed… In fact, the authors are excluding articles containing the keyword “Biomechanical” as indicated (very good) in the keywords of this manuscript.  According to this, I think that the authors do not use a good search strategy, both in the use of combinations in the non-existent use of booleans markers (AND, OR). For this reason, the authors have an excessive number of articles in the first two phases of screening (identification and selection). Can't work with 7000 papers!!! Not even to read the title! The authors properly exclude articles related to triathlon, since it would be good in this strategy to combine NOT Triathl*

In my point of view, it is the biggest methodological problem in this article.

The authors have to perform and submit a more expert search strategy and resubmit the flowchart. Thus, the reviewer (or reader) will be able to verify that after searching with this strategy in the databases used by the authors, the same articles that they report are listed.

The authors claim to include review articles. Have they included any? I have not been able to see any… it would be a strong aspect of this manuscript.

Lost reference L89 (Newcastle-Ottawa Scale).

Please submit the qualitative analysis as an appendix.

In my opinion, both figure 1 and the first paragraph of the results section should go in the method section.

Fixed W´ç33deeeer in L101

According to: "Since two studies were conducted on various swimming techniques they were included in several categories, reason why the total number of papers does not match the sum of the categories partial number. In addition, the study that evaluated the 200 m medley was included in the front crawl technique group." Do not the authors believe that would be a reason for exclusion? Since the same swimmers are contributing results in one style and in another… it is a clear bias.

Regarding table 1, it has no description in its title.

In addition, I consider that the anthropometric variables should be grouped by family for a better understanding of the table. (e. g., basic measurements such as height, wingspan, sitting height, etc.; girths, etc.) Several tables could also be presented grouping by the previous families, as analyzed in figure 2. I think that when it comes to reading the results can be better understood by the reader.

If the authors have followed the PRISMA guide as they say in the methodology, I miss basic sections such as study bias, research bias, etc. and limitations or future lines of research.

In the conclusion, the authors state that no relationship can be established regarding the breaststroke, backstroke and butterfly strokes. In my opinion, to reach this conclusion, I would exclude them from this manuscript, in order to focus conclusions on the front crawl style. For a reader, it is not pleasant to read science to come to that conclusion. I repeat, perhaps it is more scientifically appropriate to mark it in future lines of research.

From the conclusion, I only kept the second half, although the authors state again: Nevertheless, there were few variables common to the studies that 303 analyzed the butterfly, backstroke and breaststroke techniques. Repetitive.

I think this manuscript is very interesting, as long as the authors can fixed the aspects mentioned above.

Reviewer 3 Report

The article is pertinent and well written. 

The topic is of interest. The purpose of the review was to summarize the anthropometric characteristics which influence performance of young swimmers. The topic is of interest, relevant in the field of sports medicine. The conclusions are pertinent and supported by the study.  Most references are from the last 20 years and appropriate for the present study. The table and 2 figures are fine. 

I have only minor comments for improvement:

- please correct line 101,

- remove table 1 generic description.

What is the order of the studies based on? One might expect to be chronologic or by topic yet I do not see that here. 

Thank you for this opportunity.

Sincerely,

Reviewer 4 Report

Interesting and useful manuscript for scientists and coaches involved in swimming and other swimming sports, aiming to evaluate the published evidence and summarize the state of the art on the relationship between anthropometric characteristics, biomechanical variables and performance at butterfly, backstroke, breaststroke and front crawl in 9 to 18 years old swimmers.

Swimming is one of the most widespread sports, practiced recreationally or competitively by huge numbers of people of all ages. Swimming is also given a lot of attention in the scientific literature. Unfortunately, there are still publications containing methodological errors resulting from the wrong selection of participants for research, wrong research protocols or, finally, the use of inaccurate measurement tools.

In this work, the authors analyzed a large number of available publications in the field of swimming, selected several dozen works, on the basis of which they provided concise information on anthropometric and biomechanical indicators and their relationships with swimming performance.

In this work, we are dealing with very well planned and implemented research, the methods are correctly described with many details, with a very rich literature, which confirms a good understanding of the authors of the area and subject of the research undertaken. This makes this article useful for swimming scientists and coaches. A very strong point of this manuscript is the discussion in which the authors refer to the presented anthropometric parameters and explain their impact on performance in particular swimming techniques. All topics discussed in the discussion are related to the issues presented in the analysis, they are accurate and supported by relevant literature.

I believe that the work is worthy of publication and in my opinion it does not require any corrections. My comments relate only to the technical shortcomings: It is worth considering removing from keywords those included in the title of the work. On page 13, lines 101 and 110, incorrect entries should be corrected.

Round 2

Reviewer 2 Report

Good job.